# Nerve Conduction Studies of Dorsal Sural Nerve: Normative Data and Its Potential Application in ATTRv Pre-Symptomatic Subjects

**DOI:** 10.3390/brainsci12081037

**Published:** 2022-08-04

**Authors:** Marco Luigetti, Valeria Guglielmino, Marina Romozzi, Angela Romano, Andrea Di Paolantonio, Giulia Bisogni, Eleonora Sabatelli, Anna Modoni, Mario Sabatelli, Serenella Servidei, Mauro Lo Monaco

**Affiliations:** 1Fondazione Policlinico Universitario A. Gemelli IRCCS, UOC Neurologia, 00168 Rome, Italy; 2Dipartimento Di Neuroscienze, Università Cattolica del Sacro Cuore, 00168 Rome, Italy; 3Centro Clinico NEMO, Fondazione Policlinico Universitario A. Gemelli IRCCS, 00168 Roma, Italy; 4Fondazione Policlinico Universitario A. Gemelli IRCCS, UOC Neurofisiopatologia, 00168 Rome, Italy; 5MiA Onlus (“Miotonici in Associazione”), 80055 Portici, Italy

**Keywords:** ATTRv, dorsal sural nerve, monitoring, pre-symptomatic carrier

## Abstract

The objective of the study is to provide age-related normative values for dorsal sural nerve (DSN) and to analyse its application during follow-up of hereditary transthyretin amyloidosis (ATTRv) pre-symptomatic subjects. We consecutively recruited ATTRv pre-symptomatic carriers in which clinical examination, cardiological evaluation, and nerve conduction studies of the sural nerve and DSN were performed. To provide normative data of DSN, neurophysiologic parameters from healthy controls referred to our service were entered into linear regression analyses to check the relative influence of age and height. A correction grid was then derived. We collected 231 healthy subjects: the mean DSN sensory nerve action potential (SNAP) amplitude was 9.99 ± 5.48 μV; the mean conduction velocity was 49.01 ± 5.31 m/s. Significant correlations were found between age and height with DSN SNAP amplitude. Fifteen ATTRv pre-symptomatic carriers were examined. Sural nerve NCS were normal in 12/15 and revealed low/borderline values in three subjects. Considering our correction grid, we found an abnormal DNS amplitude in 9/15 subjects and low/borderline values in 2/15. In ATTRv, early detection of peripheral nerve damage is crucial to start a disease-modifying treatment. DSN may be easily and reliably included in the routine neurophysiological follow-up of ATTRv pre-symptomatic subjects.

## 1. Introduction

Hereditary transthyretin amyloidosis (ATTRv; v for “variant”) is an autosomal dominant, progressive, and life-threatening disorder caused by mutations in the transthyretin (TTR) gene [1,2]. The disease results from an extracellular deposition of amyloid fibrils in a variety of organs, leading to a multisystemic condition with a prevalent involvement of the peripheral nervous system (transthyretin amyloid polyneuropathy, ATTR-PN) and heart (transthyretin amyloid cardiomyopathy, ATTR-CM), but kidney, ocular vitreous, liver, and gastrointestinal tract may also be involved [3,4,5,6]. In the past decades, significant advances have been achieved in the treatment of ATTRv amyloidosis as several therapies with the potential to delay the disease progression have emerged [7,8,9,10], especially if started early during the course of the illness [11]. Depending on the geographic areas, a wide variation in age at onset and clinical presentation of the disease is described. Patients from endemic areas have an early-onset disease with initial involvement of small nerve fibers and frequent autonomic dysfunction, while in non-endemic areas, patients present more often with a late-onset progressive axonal polyneuropathy and cardiac involvement but with fewer autonomic symptoms [3,12].

Nowadays, TTR gene sequencing represents the diagnostic gold standard. However, the diagnosis can be confirmed by the demonstration of amyloid deposits in biopsy specimens [13].

Diagnostic techniques, along with biomarker and genetic testing, could detect the disease earlier and allow the identification of pre-symptomatic carriers [14]. This evidence highlights the importance of identification and monitoring of asymptomatic carriers of TTR mutation and the early detection of symptoms when they first occur [14].

The diagnostic techniques used in the follow-up of TTR mutation carriers include neurophysiological assessments with nerve conduction studies (NCS) [14]. The sural nerve is the sensory nerve in the lower limb, which is routinely examined in patients with peripheral neuropathies because it is accessible for nerve biopsy, allowing a correlation between pathological and neurophysiological findings [15]. However, it is well known that this site is often proximal to the sites affected by the earliest signs of distal polyneuropathies [15]. Indeed, the investigation of dorsal sural nerves (DSN) is an approach that could overcome this limitation, exploring one of the most distal sensory nerve branches [16]. NCS of DNS, in the routine neurophysiological diagnostic process of polyneuropathies, may allow early detection of peripheral nerve damage. However, DSN may vary considerably depending on the age of the subjects, and currently, there are no available normative values [16].

In this paper, we performed a study on a large cohort of healthy adults over an extended age range to provide age-related normative values for DSN and we analysed its application during follow-up of ATTRv pre-symptomatic subjects.

## 2. Materials and Methods

### 2.1. Study Population

We consecutively recruited ATTRv pre-symptomatic carriers aged >18 years referred to the Neurophysiology Service of Fondazione Policlinico Universitario A. Gemelli IRCCS in Rome. Enrolled subjects gave written informed consent to participate in the study. The study conforms to the ethical guidelines of the 1975 Declaration of Helsinki (6th revision, 2008). The study was approved by Ethic Committee of Fondazione Policlinico A. Gemelli IRCSS (Prot. ID 4108).

As a control population, we considered subjects with no history or risk factors for neuropathies (such as diabetes mellitus, systemic diseases, alcohol abuse, or use of drugs that can cause neuropathy) and neither signs nor symptoms suggestive of neuropathy (no numbness, tingling or burning sensation in the lower limbs and normal muscle strength) referred to our service from 2005 to 2015 because of several reasons such as radiculopathy, low back pain, carpal tunnel syndrome or other compressive neuropathies.

In pre-symptomatic subjects, we performed a clinical examination, cardiological evaluation, and nerve conduction studies of the sural nerve, as previously described [15,17].

### 2.2. Nerve Conduction Studies

DSN was examined with a standard antidromic technique. The surface bar recording electrodes were placed over the lateral dorsal surface of the foot, with the distal electrode at the origin of digits 4 and 5 and the proximal active electrode 3 cm from the distal electrode, the stimulation site was posterior to the lateral malleolus, directly over the sural nerve, with the cathode placed 14 cm from the proximal recording electrode, the ground electrode was placed on the dorsum of the foot equidistant between the recording and stimulating electrodes [16,18]. In an attempt to correlate neurophysiological findings obtained in the sural and DSN nerves, the ratio between SNAP of the sural nerve and DSN was also calculated.

### 2.3. Data Analysis

Statistical analysis was performed as previously described for the sural nerve [15]. Correlation coefficients were obtained in order to assess the relationship between age, height, and neurophysiological parameters. Afterwards, neurophysiologic parameters were entered into linear regression analyses in order to check the relative influence of age and height. The effects of age and height were also studied after various transformations (logarithmic, quadratic, inverted, subtraction) and the most effective transformation in reducing the residual variance was adopted. An adjusted score was calculated for each subject by adding or subtracting the contribution of the concomitant variables from the original parameter. A correction grid was then derived to allow immediate adjustment of the raw performance of newly tested individuals. After ranking the adjusted scores from worst to the best performance, we also computed inner and outer tolerance limits. Parametric techniques cannot be used with adjusted scores, whereas nonparametric techniques only require that the scores possess ordinal scale properties. Therefore, we determined nonparametric tolerance limits according to Wilks’ method [19], which allows to determine an inner tolerance limit that can be assumed as a cut-off and an outer tolerance limit that can be assumed as the lowest normal value; the values included between the inner and outer tolerance limits can be considered as “borderline values”.

The cohort was described using descriptive statistics techniques in its clinical and demographic features. Quantitative variables were described using mean, median, range, and standard deviation (SD). The distribution of each numerical variable was checked with Shapiro–Wilk test. Comparisons were performed with Mann–Whitney or *t*-test as appropriate. The linear correlation between two variables was tested using Spearman *rho*. The significance level was set at *p* < 0.05.

## 3. Results

### 3.1. Healthy Controls

We collected 231 healthy subjects. The mean age of the sample was 51.38 years ± 17.32 (range 8–92); the mean height was 168.55 ± 9.32 cm (range 133–190). The mean height of the female subjects was 163 ± 5.80 cm; the mean height of the male subjects was 174 ± 8.77. There was a statistically significant difference in the average height between male and female subjects (*p* < 0.001) (the demographic data of healthy subjects are available in Appendix A). The mean sural SNAP amplitude was 21.66 ± 10.30 μV (range 4.7–68.0); the mean conduction velocity was 54.98 ± 5.33 m/s (range 42–69). The mean DSN SNAP amplitude was 9.99 ± 5.48 μV (range 0.6–47); the mean conduction velocity was 49.01 ± 5.31 m/s (range 37–69). The mean sural/DNS amplitude ratio was 2.37 ± 0.90 (range 1.0–3.51). There was a statistically significant difference between male and female subjects for DSN SNAP amplitude [respectively, 8.45 ± 4.25 vs. 11.51 ± 6.17; *p* < 0.001] and not for NCV [respectively, 49.52 ± 5.81 vs. 48.50 ± 4.72; *p* = 0.524].

### 3.2. Linear Regression Model

Significant correlations were found between age and DSN SNAP amplitude (*rho* = −0.418; *p* < 0.001) and not with NCV (*rho* = −0.108; *p* = 0.102). Analogously, height displayed significant correlation with SNAP amplitude (*rho* = −0.140; *p* = 0.033) and not with NCV (*rho* = −0.030; *p* = 0.647). For this reason, in order to exclude collinearity effects that may make regression models less reliable, we chose to enter into the models only the variable displaying the strongest correlation with the dependent variable together with gender.

The lower residual variance among the models predicting DSN SNAP with age as a dependent variable was obtained using the logarithmic transformation of age [log(100 − age)]. The final model was:SNAP amplitude = 8.235 + 5.384 × [log(100 − age) − 3.80] + 3.529 × gender (F = 1; M = 0)

This model accounted for about 25% of total variance (adjR^2^ = 0.252); 3.80 is the arithmetical mean of log(100 − age). Table 1 reports the correction grid for age with a five-year step; corrected SNAP can be obtained by summing the measured SNAP to the corresponding correction factor. The inner limit of the tolerance limit, corresponding to the cut-off score, was 7.357; the outer limit, corresponding to the lowest normal value, was 9.112. Corrected SNAP amplitude comprised between these values should be considered “borderline”.

### 3.3. ATTRv Pre-Symptomatic Carriers

Fifteen ATTRv pre-symptomatic carriers were examined. Demographic and neurophysiological data are summarised in Table 2. Clinical examination and cardiological evaluation were unremarkable in all subjects. Sural nerve NCS were normal in 12/15 and revealed low/borderline values in three (#7, #11, and #12).

Considering our correction grid, we found an abnormal DNS amplitude in 9/15 subjects (60%) and low/borderline values in 2/15 (#2 and #12). An increased sural/DNS amplitude ratio, defined as >4.17 (mean values +2 SD), confirming this data, was found in 6/9 (66%). In two subjects (#7 and #11), the concomitant presence of low/borderline values of sural nerve amplitude could explain this result.

If we divided the subjects into two groups, considering age at examination accordingly to predictive age of disease onset (PADO), defined as ten years before the onset of the affected relative, we found a significant difference regarding abnormal DSN SNAP distribution (5/10 subjects before PADO and 4/5 after PADO, *p* = 0.017).

## 4. Discussion

In ATTRv, early detection of peripheral nerve damage is crucial to start a disease-modifying treatment [11]. ATTRv is generally a length-dependent polyneuropathy, and the most distal sensory fibers are affected first [12]. However, classical neurophysiological follow-up of pre-symptomatic carriers is generally limited to the sural nerve that does not include the most distal sensory fibers [14]. We examined our cohort of ATTRv pre-symptomatic carriers and also DSN, and we found abnormal values, according to our normative data, in 60% of the subjects. DSN can be accurately and easily tested by using standard electromyography equipment, as already reported in a few reports and confirmed by this normative study [16]. The DSN study was successfully performed in all the healthy subjects examined in our series. A significant age-related decrease in DSN SNAP amplitude was found, which agrees with what has been observed previously in sural nerve studies [15].

We suggested to also evaluate in ATTRv sural/DSN amplitude ratio that, if increased, can confirm length-dependent nerve damage: distal nerve fibers may be firstly involved not only in early-onset ATTRv but also in late-onset, as recently suggested by studies about intra-epidermal nerve fibers density [20].

Furthermore, we found a significant difference in the proportion of abnormal DSN amplitude values considering PADO: this data may suggest the importance of this test as an early marker of nerve damage in age at risk.

Recently, in ATTRv, radiological or serological biomarkers have been proposed to follow up pre-symptomatic carriers in order to detect early signs of the disease [21,22,23]. The diagnosis of conversion of symptomatic ATTRv amyloidosis relies mainly on clinical symptoms, although the change from baseline in NCS represents a criterion for conversion [14]. NCS can significantly change 1–2 years before the development of ATTR-PN symptoms [24]. Nevertheless, the use of composite neurophysiological scores is considered more adequate than individual NCS studies to assess disease progression [24].

A limitation of the study could be due to the great variability of DSN values, which we have tried to overcome by providing normative data. Another limitation is the small number of pre-symptomatic carriers examined in the study.

## 5. Conclusions

We can conclude that DSN may be easily and reliably included in the routine neurophysiological follow-up of ATTRv pre-symptomatic subjects. The normative data we have described could serve as reference values. DSN could also be employed in a neurophysiological sum score to facilitate the early detection of conversion.

Abnormal values may lead clinicians to start a disease-modifying therapy; on the other hand, normal, or low/borderline, values can be observed during regular follow-up (as commonly used for sural nerve) in order to detect a pathological amplitude decrease. Longitudinal studies on a larger cohort and follow-up studies will clarify the significance of our result.

## Figures and Tables

**Table 1 brainsci-12-01037-t001:** Correction grid for sensory nerve action potential of the dorsal sural nerve.

Age	Male	Female
20	−3.13	−6.66
25	−2.79	−6.34
30	−2.42	−5.95
35	−2.02	−5.55
40	−1.59	−5.12
45	−1.12	−4.65
50	−0.60	−4.13
55	−0.04	−3.57
60	+0.60	−2.93
65	+1.32	−2.21
70	+2.15	−1.38

**Table 2 brainsci-12-01037-t002:** Demographic findings and raw neurophysiological data of pre-symptomatic ATTRv carriers.

Car	TTR Variant	Age	Sex	PADO	Sural Nerve SNAP (μV)	Sural Nerve CV (m/Sec)	DSN SNAP (μV)	DSN CV (m/Sec)	Sural/DSN Amplitude R
#1	V30M	42	F	54	29.8	65	19.9	51	1.50
#2	V30M	39	F	54	31.0	54	12.5	47	2.48
#3	V30M	40	M	54	43.3	67	35.1	66	1.23
#4	F64L	73	F	51	16.6	67	3.7	51	4.49
#5	F64L	43	F	51	26.6	55	4.5	55	5.91
#6	F64L	49	M	51	22.1	57	10.0	41	2.21
#7	V30M	42	F	50	17.0	54	8.7	56	1.95
#8	V30M	49	M	65	26.3	55	6.1	42	4.31
#9	V30M	65	M	50	16.9	59	8.8	55	1.92
#10	V30M	69	M	50	6.3	61	3.6	53	1.75
#11	V30M	45	F	56	16.3	64	4.2	49	3.88
#12	V30M	43	M	56	12.8	45	8.9	43	1.44
#13	V30M	55	F	62	38.9	66	7.6	56	5.12
#14	V30M	72	M	67	11.1	51	2.2	45	5.05
#15	V30M	68	M	67	11.2	50	2.5	46	4.48

Legend: Car, carriers; TTR, transthyretin; PADO, predictive age of disease onset; SNAP, sensory nerve action potential; CV, conduction velocity; DSN, dorsal sural nerve; R, ratio. Abnormal values are written in italics. Patients #1, #2, #3 come from the same kindred as well as patients #4, #5, #6, patients # 9 and #10, patients #11 and #12, patients #14 and #15.

## Data Availability

The data that support the findings of this study are available from the corresponding author upon reasonable request.

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
