# Peer review of "Nerve Conduction Studies of Dorsal Sural Nerve: Normative Data and Its Potential Application in ATTRv Pre-Symptomatic Subjects"

_brainsci, 2022, doi:10.3390/brainsci12081037_

Round 1
Reviewer 1 Report
In this manuscript, Luigetti t al provide the normative values for dorsal sural nerve (DNS) response and derived a correction grid based on age and gender. Using the corrected values, they found an abnormal DNS amplitude in 9/15 subjects and increased sural/DNS amplitude ratio in 6/9 subjects with presymptomatic hereditary TTR amyloidosis. The authors had used similar methodology and models for deriving corrected sural sensory response in the past. Overall, the manuscript is well written. Interpretations are straightforward. Here are some questions/comments:
- Without using the correction grid, did the authors observe abnormal DNS amplitude or sural/DNS amplitude just based on normal uncorrected values for each 10 step age group?
- In Table 2, it is assumed that both sural and DNS amplitudes listed are the corrected values, though not clearly stated. In pt#10, the sural amplitude was the lowest and would be considered abnormal based on the authors’ 2012 publication. Any reason why it was not even considered low or borderline here?
- The authors stated that DSN amplitude is influenced by gender and height. Is there a difference in the height between males and females, and if so, could this explain the influence of gender on normal DNS amplitude?
Author Response
Please see PDF attached.

Reviewer 2 Report
The study utilised dorsal sural nerve conduction studies for the diagnosis of ATTRv amyloidosis with polyneuropathy.
1. To rewrite the research gap and limitations of previous studies. How does the application of Dorsal Sural Nerve overcome the limitation of previous study / studies?
Line 55-63:
The sural nerve is the sensory nerve in the lower limb which is routinely examined in patients with peripheral neuropathies, because it is accessible for nerve biopsy, allowing a correlation between pathological and neurophysiological findings [15]. However, it is well known that this site is often proximal to the sites affected by the earliest signs of distal polyneuropathies [15]. The investigation of dorsal sural nerves (DSN) is an approach that could overcome this limitation, exploring one of the most distal sensory nerve branches [16]. Though, an age-related variation in DSN sensory nerve action potential (SNAP) amplitude and nerve conduction velocity (NCV) was found in previous studies [16].
2. To expand Methodology section
(i) Study population with reference for Ethical Clearance should be first in the beginning of this section - inclusion and exclusion criteria, groupings and age of subjects
(ii) Nerve conduction studies
(iii) Data analysis
3. To include a section of Linear regression model for normative reference values in the Results section. Also to include a Supplementary Material / Data (an Excel file) of the age and height of the subject for the calculation of corresponding normative values.
4. What are the limitations of your study?
5. Sentence construction and rephrasing are needed for better understanding and to improve readibility. Kindly go through the manuscript
Author Response
Please see PDF attached.
